# Clinicopathological Profiles Associated with Discordant *RAS* Mutational Status between Liquid and Tissue Biopsies in a Real-World Cohort of Metastatic Colorectal Cancer

**DOI:** 10.3390/cancers15143578

**Published:** 2023-07-12

**Authors:** Elena Brozos-Vázquez, Ramón Manuel Lago-Lestón, Marta Covela, Juan de la Cámara Gómez, Ana Fernández-Montes, Sonia Candamio, Yolanda Vidal, Francisca Vázquez, Alicia Abalo, Rosa López, Cristina Blanco, Laura Muinelo-Romay, Isabel Ferreirós-Vidal, Rafael López-López

**Affiliations:** 1Translational Medical Oncology Group, Oncomet, University Hospital of Santiago de Compostela (CHUS), 15706 Santiago de Compostela, Spain; elenambrozosv@hotmail.com (E.B.-V.); ramonlago.oncomet@gmail.com (R.M.L.-L.); sonia.candamio.folgar@sergas.es (S.C.); yolanda.vidal.insua@sergas.es (Y.V.); franciscavazquezrivera@yahoo.com (F.V.); alicia.abalo.pineiro@sergas.es (A.A.); lopezrodriguezrosa@gmail.com (R.L.); cblancofreire@gmail.com (C.B.); lmuirom@gmail.com (L.M.-R.); 2Liquid Biopsy Unit, Health Research Institute of Santiago de Compostela (IDIS), 15706 Santiago de Compostela, Spain; 3Department of Medical Oncology, Lucus Augusti University Hospital of Lugo (CHULA), 27003 Lugo, Spain; martacovela@gmail.com; 4Department of Medical Oncology, University Hospital of Ferrol (CHUF), 15405 Ferrol, Spain; jcamarakq@gmail.com; 5Department of Medical Oncology, University Hospital Complex of Ourense (CHUO), 32005 Ourense, Spain; afm1003@hotmail.com; 6Centro de Investigación Biomédica en Red Cáncer (CIBERONC), 28029 Madrid, Spain

**Keywords:** liquid biopsy, colorectal cancer, metastatic colorectal cancer, mCRC, RAS, ctDNA, anti-EGFR therapy

## Abstract

**Simple Summary:**

The evaluation of *RAS* mutations from plasma circulating tumour DNA (ctDNA) has emerged as an efficient approach to guide therapeutic decisions in the treatment of metastatic colorectal cancer (mCRC). However, disagreements about the *RAS* mutational status in tests using liquid and tumour tissue biopsies can lead to misinterpretation of the tumour genotype and compromise effective therapy. We used digital droplet PCR (ddPCR) technology and logistic regression models to decipher the clinicopathological profiles that commonly influence the levels and detection of plasmatic *RAS* mutations and are associated with discordant assays. The absence of liver metastases and the resection of the primary tumour were associated with reduced ctDNA levels and low percentages of positive agreement between tissue and ctDNA tests, predominantly when the mCRC originated in the right colon and rectum. Thus, ctDNA assays reporting undetected *RAS* mutations in these patients should be taken prudently, and further investigations should be considered before any decision about treatment.

**Abstract:**

We aimed to identify common mCRC profiles associated with a discordant mutational status of *RAS* between the standard of care (SoC) tumour tissue tests and ctDNA tests to understand ctDNA detection and improve treatment responses. This was a multicentre, retrospective and prospective study. A total of 366 Spanish mCRC patients were independently recruited. BEAMing ddPCR technology was employed to detect ctDNA *RAS* mutations, and logistic regression analyses were performed to investigate clinicopathological factors associated with discordance. The highest concordance ratios were observed in profiles with multiple metastatic sites when the liver was present (89.7%; 95% CI 84.8–93.2), profiles with synchronous disease without primary tumour resection (90.2%; 95% CI 83.6–94.3) and profiles with mCRC originating in the left colon (91.3%; 95% CI 85.0–95.0). Metachronous disease originating in the right colon (OR = 6.1; 95% CI 1.7–26.5; *p*-value = 0.006) or rectum (OR = 5.0; 95% CI 1.5–17.8; *p*-value = 0.009) showed the highest probability of discrepancies. Primary tumour resection and a higher frequency of single metastases in the peritoneum or lungs in these patients were associated with reduced plasmatic mutation allele fractions (MAFs) and an increased probability of showing false-negative genotypes. Additional testing of patients with mCRC originating in the right colon or rectum with a single non-mutated ctDNA test is advised before the choice of therapy.

## 1. Introduction

Colorectal cancer (CRC) is one of the most frequently occurring solid cancers worldwide, with a higher incidence in developed countries associated with lifestyle risk factors, such as diet and lack of exercise [1]. The prevalence patterns are generally similar in men and women, but an alarming increased rate has lately been observed in adults younger than 55 years old [2,3,4]. Notably, it is the second leading cause of cancer-related death in Europe [5], and Spain has been determined to be one of the European countries with the highest incidence and mortality rates [1].

The aetiology, physiopathology and molecular characteristics of CRC are highly heterogeneous. Adenocarcinoma has the potential to originate in any location in the colon or rectum, and the molecular features of right-sided colon cancers are different compared with left-sided colon and rectal cancers [6]. These differences in CRC aetiology have a key role in the metastatic settings of the disease. The burden and dissemination patterns [7,8] of mCRC have been shown to significantly depend on the location of the primary tumour. As a consequence of the great heterogeneity in mCRC molecular and clinicopathological characteristics, systemic therapy for mCRC patients must be personalised and dependent on the disease-specific predictive markers. 

The mutational status of the rat sarcoma viral oncogene homolog (*RAS*) family of genes—and, more recently, the tumour sidedness—are the current mCRC biomarkers used to predict the targeted therapy response in clinical practice [6]. Cetuximab and panitumumab are monoclonal antibodies against the epidermal growth factor receptor (*EGFR*) that have been demonstrated to improve outcomes in mCRC patients with wild-type (WT) status compared to the anti-VEGF used as the first line of treatment [9,10,11], particularly in those patients whose primary tumour originated in the left side of the colon [12,13,14,15,16,17]. However, approximately 40% of CRCs show activating missense mutations in the Kirsten *RAS* (*KRAS)* gene and 3–5% in the neuroblastoma *RAS (NRAS)* [18], representing an early event in the carcinogenesis of mCRC [19]. Therefore, around half of mCRC cases show resistance to EGFR-targeted therapy. Hence, current guidelines for the clinical management of mCRC recommend the evaluation of the *RAS* mutational status, especially exons 2, 3 and 4 of the *KRAS* and *NRAS* genes, before the administration of *EGFR* inhibitors [20,21,22].

The standard of care (SoC) procedure in clinical practice is to evaluate the presence of somatic *KRAS* and *NRAS* mutations with a solid tumour specimen from surgical or biopsy samples at the diagnosis of the metastatic disease [23], commonly by using mutation-specific sequencing-based technologies or, more recently, next-generation sequencing (NGS) approaches [24]. However, the evaluation of *RAS* mutations using circulating tumour DNA (ctDNA) from minimal-invasive blood-based liquid biopsies has been established as a valuable approach to stratify and monitor mCRC patients in real time throughout the course of the disease [25], guiding clinical decisions more efficiently [26,27,28,29]. Also, in other tumour types, such as NSCLC, *EGFR* genotyping through ctDNA assessment is well-accepted, especially when invasive procedures may be risky or contraindicated [30,31]. 

Previous studies [32,33,34] have shown high correlations, with concordance ratios above 85%, between routine sequencing procedures performed with solid tumour tissue and the OncoBEAM^®^ *RAS* CRC, a technology that uses BEAMing digital droplet PCR and has been approved by the European Commission as a tool for the in vitro diagnosis of *RAS* mutations [35]. Prospective studies including larger cohorts of patients have already elucidated some of the frequent causes of discordance between tumour tissue and plasma ctDNA analysis, highlighting the impact of clinical features on the detection of plasmatic ctDNA. Small amounts of plasmatic ctDNA in mCRC patients with resection of the primary tumour or absence of liver metastases led to undetectable *RAS* mutations in plasma and inconclusive results concerning concordance [36,37,38,39]. In addition, ctDNA *RAS* mutant allele fraction (MAF) in the mCRC population with exclusive lung metastatic disease was frequently found to be lower than in patients with at least one liver metastasis, which led to false-negative results in the ctDNA-based tests [40,41]. 

Liquid biopsy tests reporting false-negative results are expected to have an impact on the choice of therapy. Despite broad scientific knowledge about the disease conditions potentially leading to inconclusive results, difficulties exist in the day-to-day medical care to interpret discordances between liquid biopsy tests and SoC procedures performed in tumour tissue, thereby compromising the selection of the best candidates to receive anti-*EGFR* therapy. 

A better understanding of the clinicopathological characteristics that predominantly impact the molecular detection of *RAS* mutations in mCRC is still needed to identify properly the mCRC subpopulations susceptible to presenting misleading ctDNA results. Enhanced patient stratification will improve tailored therapy and the definitive establishment of ctDNA testing in the clinical routine.

## 2. Materials and Methods

### 2.1. Study Design and Patients 

This is a multicentre, retrospective and prospective observational study. A total of 366 unrelated mCRC patients were independently recruited from November 2015 to July 2019 in four hospitals in Galicia, northwest Spain: 233 patients recruited from the University Hospital of Santiago de Compostela, 67 from the Lucus Augusti University Hospital of Lugo, 38 from the University Hospital of Ferrol and 28 patients from the University Hospital of Ourense. 

Patients had been previously diagnosed with Stage I, Stage II (IIA, IIB, IIC), Stage III (IIIA, IIIB, IIIC) or Stage IV (IVA, IVB and IVC) CRC following the recommendations of the American Joint Committee on Cancer (AJCC) TNM Staging System for Colon Cancer [42] published at the NCCN Guidelines for Colon Cancer [43]. All patients were selected to have been histologically and radiologically diagnosed with Stage IV metastatic colorectal cancer at the time of enrolment in this study.

All patients had at least one blood draw taken at the metastatic stage. In most cases (79%), blood biopsies were obtained within four months of the diagnosis of metastatic disease. The average time interval from the collection of the solid tumour tissue biopsy to the liquid blood biopsy was less than 3 months (70 days on average) for patients diagnosed with synchronous metastatic disease and a year and a half (535 days on average) for patients diagnosed with recurrent metachronous metastatic disease. 

Information regarding the mutational status of *RAS* in the tissue tumour sample was available in 83% of the patients. On the other hand, *RAS* genotyping by BEAMing ddPCR was achieved in 99.5% of the plasma blood samples.

The regional ethics committee (Comité Ético de Investigación Clínica de Galicia) approved the study (ref 2015/746), and written informed consent was obtained from all patients.

### 2.2. Procedures

Peripheral blood samples were collected in 10 mL Cell-Free DNA STRECK BCT^®^ tubes (Streck, Nebraska, United States). Two-step centrifugation was carried out to isolate the plasma from the whole blood. Blood samples were centrifuged for 10 min at 1600× *g* at room temperature. The supernatant was collected and centrifuged for 15 min at 6000× *g* to remove the remaining cells. Plasma samples were stored in aliquots at −80 °C until further analyses. Plasma cell-free DNA (cfDNA) was isolated from 5 mL of peripheral blood using QIAamp Circulating Nucleic Acid Kit (Qiagen, Hilden, Germany) and quantified with a Qubit dsDNA HS Assay Kit (Thermo Fisher Scientific, Massachusetts, United States). *RAS* genotyping was performed using BEAMing Digital Dropplet PCR (ddPCR) Technology (OncoBEAM ^TM^ RAS CRC CEIVD kit, Sysmex Inostics GmbH, Hamburg, Germany), which detected 34 mutations in the codons 12, 13, 59, 61, 117, and 146 of the *KRAS* and *NRAS* genes, and compared the *RAS* mutational status in formalin-fixed and paraffin-embedded (FFPE) tissue sections derived from primary tumours or from metastasized tumour tissue, which had been previously genotyped according to the procedures validated in each of the hospitals of the patients recruited: Therascreen KRAS RGQ PCR kit (Qiagen); COBAS KRAS mutation test (Roche Diagnostics, Rotkreuz, Switzerland) or pyrosequencing (PyroMark Q24, Qiagen) testing.

### 2.3. Statistical Analysis

Statistical analyses were performed with clinically predefined mCRC subpopulations.

Concordance between plasma- and tumour-tissue-based analyses were determined using a Kappa statistic (kappa) with a 95% confidence interval (CI). Overall concordance, sensitivity (positive percent agreement), and specificity (negative percent agreement) between plasma- and tissue-based analyses and their 95% CIs were also calculated. 

Binary logistic regression was used to model the relationship between the mCRC clinicopathological characteristics and the *RAS* mutational status (discordant vs. concordant) taken as the dichotomous dependent variable. The main outcome of the analysis was the odds of discordance versus concordance between SoC and ctDNA tests. We used a generalized linear model and a logit function to estimate the odds ratios (ORs) of discordance and their 95% confidence intervals (CIs), and to calculate the probability of association with the site of the primary tumour, the resection of the primary tumour, the metastatic site and the number of metastatic sites taken as the independent variables. Patients with mCRC originating in the left colon, without primary tumour resection, with liver metastases, lymph node metastases, lung metastases or peritoneal metastases, and with multiple metastatic sites were used as the reference for the independent variables. 

We used univariable logistic regression to estimate the effect of each of the clinicopathological characteristics on the odds of discordance, individually. Multivariable logistic regression models were also used to estimate the joint effect or interaction of a set of clinicopathological variables on the probability that the outcome happened. 

Finally, univariable logistic regression analyses in mCRC subpopulations stratified by the location of the primary tumour (left colon, rectum and right colon) were undertaken to model the effect of the resection of the primary tumour, the metastatic site and the number of metastatic sites on the odds of discordance occurring. 

The median and interquartile range (25–75%; IQR) of the cfDNA levels and MAFs distributions were calculated for the different mCRC subpopulations. The Mann–Whitney and the Kruskal–Wallis non-parametric approaches were applied for the statistical comparisons between two or more populations, respectively.

All statistical tests were performed in R version 3.6.3, GraphPad Prism 6 and the VassarStats: Website for Statistical Computation. Odds ratios ≥ 2 and *p*-values ≤ 0.05 were considered statistically significant.

## 3. Results

### 3.1. Cohort Characteristics

A total of 366 patients were recruited from four hospitals in Galicia, northwest Spain. At the time of the inclusion in the study, all patients had been diagnosed with mCRC. Disease staging was based on the histopathology of biopsies or surgical specimens and on imaging diagnoses. The baseline characteristics of patients are shown in Table 1. The majority (71.6%) of mCRC patients were male and the average age was 66 ± 11 years old. The primary tumour was located in the left side of the colon in 42% of patients, in the right side of the colon in 22% of patients, in the rectum in 31% of patients, and was of mixed origin in 5% of patients. Considering the timing of the diagnosis of metastatic disease, 70% of patients had been diagnosed with synchronous primary and metastatic tumours while 29% of patients presented a metachronous onset of the metastatic disease. About half of the patients showed only a single location of metastases at the diagnosis of mCRC. In these cases, the liver was the most frequent metastatic location (61.8%), followed by the lung (14.2%) and the peritoneum (13.2%). In addition, the liver was also the most common metastatic site among patients presenting metastases at more than one anatomical location. Almost 70% of these patients had been diagnosed with metastases in the liver and in other anatomical locations.

All mCRC patients had at least one blood liquid biopsy taken in the metastatic stage. At blood biopsy collection, 60% of patients had undergone surgical resection of the primary tumour and 12% had undergone resection of metastasized tumour lesion due to any local complication. A total of 69% of patients were chemotherapy naïve, 22% had received adjuvant chemotherapy after primary tumour resection, and 15% had received systemic chemotherapy (+anti-VEGF in three of the cases) for palliative or life-extending care. Thirty-three percent of patients received anti-EGFR therapy after blood sample collection. 

### 3.2. Concordance of RAS Mutational Status between Analyses Based on Solid Tumour Tissue and Analysis Based on Plasmatic ctDNA

*RAS* mutations were found in 47% of solid tumour tissue samples genotyped with SoC techniques and in 49% of plasma ctDNA samples analysed by BEAMing ddPCR (Table 1). 

The overall concordance of the *RAS* mutational status between the tests in 301 mCRC samples with genotyping information in both solid tumour tissue and plasma liquid biopsies was 84.4%, CI 79.9–88.1 (Cohen’s Kappa = 0.69; 95% CI 0.61–0.77), with a sensitivity of 81.3% and a specificity of 88.5% (Table 2). The highest overall concordance ratios of the study were observed in the mCRC population with multiple metastatic sites when the liver was present (89.7%, CI 84.8–93.2), in patients with synchronous mCRC that was not previously treated with resection of the primary tumour (newly diagnosed synchronous mCRC) (90.2%, CI 83.6–94.3) and in patients with mCRC originating in the left colon (91.3%, CI 85.0–95) (Table 2).

### 3.3. Association of the mCRC Clinicopathological Features with Discordance of RAS Mutational Status between Analysis Based on Tumour Tissue and Analysis Based on Plasmatic ctDNA 

We used logistic regression analyses to decipher the clinicopathological characteristics with the most significant association with discordance in our mCRC cohort (Table 2). The metastatic pattern was a main factor predictor of the discordance between assays. In patients with metastases at more than one anatomical location, the absence of liver metastases significantly increased the probability of discordance (OR = 2.7, 95% CI [1.4–5.3]; *p*-value = 0.003). We observed that the sensitivity rates were significantly decreased among tests of patients with only one metastatic site (76.84% vs. 89.09%; *p*-value = 0.044), mainly when this site was located at the lung (63.2%) or the peritoneum (53.8%). Consistent with this observation, patients with only metastases in the lung (OR = 3.2, 95% CI [1.1–8.8]; *p*-value = 0.029) or the peritoneum (OR = 3.3, 95% CI [1.1–9.5]; *p*-value = 0.035) showed the highest probability of having a mutational status of *RAS* discordant between analyses based on tissue and on plasma. 

In addition to the absence of liver metastases, the resection of tumour tissue was also significantly associated with a lack of concordance. The probability of having discordant results between assays increased when patients had undergone the resection of tumour tissue before the collection of the blood liquid biopsy (OR = 1.9, 95% CI [1.0–3.9]; *p*-value = 0.073), mostly noticeable in profiles with metachronous mCRC (OR = 2.9, 95% CI [0.4–2.5]; *p*-value = 0.007) than in profiles with synchronous mCRC (OR = 1.5, 95% CI [0.6–3.4]; *p*-value = 0.365) (Table 2 and Appendix A). Discordance associated with metachronous mCRC populations was also dependent on the metastatic locations since multivariable logistic regression analyses showed that the probability of showing discordance was reduced in the metachronous mCRC population with liver metastases (OR = 1.9, 95% CI [0.8–4.6]; *p*-value = 0.127). 

### 3.4. Association of the mCRC Origin Site with Discordance of RAS Mutational Status between Analysis Based on Tumour Tissue and Analysis Based on Plasmatic ctDNA 

The probability of discrepancies between both assays was strongly associated with the anatomical location of the primary tumour. Patients with mCRC originating in the right colon had a likelihood of presenting discordant genotypes three times higher than patients with primary tumours located in the left colon (OR = 3.1, 95% CI [1.4–7.5]; *p*-value = 0.008). Moreover, patients with primary tumours originating in the rectum showed discordant assays twice as frequently as patients with primary tumours in the left colon (OR = 2.1, 95% CI [0.9–4.8]; *p*-value = 0.083) (Table 2). 

The probability of discrepancies between assays in both populations, the mCRC originated at the right colon (OR = 6.1, 95% CI [1.7–26.5]; *p*-value = 0.006) and the mCRC originated at the rectum (OR = 5.0, 95% CI [1.5–17.8]; *p*-value = 0.009), increased significantly among patients without liver metastases (Table 3). More specifically, the highest frequencies of discordant *RAS* status in the mCRC originating in the right colon were observed in patients with only peritoneal metastases (OR = 7.7, 95% CI [1.3–54.5]; *p*-value = 0.030), and in cases with mCRC originated in the rectum, discordant results increased in patients with only lung metastases (OR = 5.8, 95% CI [1.3–2.6]; *p*-value = 0.018).

In addition, the resection of the tumour before the collection of the blood liquid biopsy significantly increased the chances of obtaining discordant results between assays, particularly among patients with metachronous mCRC originated in the rectum (OR = 8.8, 95% CI [2.6–36]; *p*-value = 0.001) and in the right colon (OR = 2.9, 95% CI [0.6–16.2]; *p*-value = 0.187) (Table 3 and Appendix A).

These two mCRC populations, patients with metachronous mCRC originated in the rectum (56.3%) or in the right colon (54.6%), showed the lowest sensitivity ratios as well as the lowest confidence ratios to predict true plasmatic WT *RAS* genotypes, as reflected by their negative predicted value (NPV) (Table 4). Thus, compared to a 13.6% (95% CI 3.6–36.0) probability of showing a false-negative plasmatic WT *RAS* genotype in the population with metachronous mCRC originating in the left colon, this probability was 50.0% (95% CI 20.1–79.9) and 53.8% (95% CI 26.1–79.6) in cases with right-sided and rectal metachronous mCRC, respectively (Table 4).

Additionally, 12% of cases with metachronous mCRC originating in the rectum showed *RAS* mutations in plasma, despite *RAS* mutations not being observed in their tumour tissue assays, demonstrating that the plasma assays from patients with metachronous mCRC originating in the rectum showed the lowest specificity of the study as well (Table 4).

### 3.5. Impact of the mCRC Clinicopathological Characteristics on the ctDNA RAS Mutational Load 

The plasmatic *RAS* mutational load has been shown to be lower in patients without liver metastases [32,37,41] and to decrease upon primary tumour resection [34,40] or systemic chemotherapy [32]. 

The median of *RAS* MAFs in plasmatic ctDNA from 165 patients with *RAS* mutated genotypes was 0.029, IQR [0.005–0.104], ranging from a minimum level of 0.0002 to a maximum of 0.58. This wide dispersion suggested that the *RAS* MAF levels were highly variable among the mCRC patients. 

According to the above-cited studies, we observed that the ctDNA *RAS* MAFs were significantly higher in patients with at least one liver metastasis than in patients without liver metastases but with metastases at other anatomical locations (median = 0.044, IQR [0.007–0.129] vs. median = 0.011, IQR [0.003–0.039]; *p*-value < 0.001) (Figure 1a). In line with these observed ctDNA levels, fluorometric quantification of plasmatic cfDNA estimated that the mCRC subpopulation with at least one liver metastasis showed significantly higher levels of cfDNA in plasma than patients who did not have any liver metastases but had metastases at other sites (median = 36.7 ng/mL, IQR [17.7–87.3] vs. median = 12.9 ng/mL, IQR [9.8–18.9]; *p*-value < 0.001). cfDNA levels were also higher in patients who had metastases only at the liver than in patients with lung metastases alone (median = 37.1 ng/mL, IQR [20.4–80.4] vs. 13.2 ng/mL, IQR [12.5–17.2], *p*-value = 0.004) or exclusive peritoneal metastatic disease (median = 37.1 ng/mL, IQR [20.4–80.4] vs. 12.4 ng/mL, IQR [9.8–20.5], *p*-value = 0.031).

Furthermore, we observed that patients diagnosed with synchronous mCRC, who had not been treated either with tumour resection or with systemic chemotherapy at the moment of the blood biopsy extraction, had higher ctDNA *RAS* MAFs than patients with synchronous mCRC who were naïve to systemic chemotherapy but treated with resection of the primary tumour (8.9% had also been subjected to resection of metastasized tissue), (median = 0.057, IQR [0.016–0.125] vs. median = 0.025, IQR [0.003–0.079]; *p*-value = 0.010) and also significantly higher than the plasmatic *RAS* MAFs that were observed in patients with a metachronous presentation of the metastases (median = 0.009, IQR [0.003–0.028]; *p*-value < 0.001) (Figure 1b). 

Lower ctDNA *RAS* MAF medians were observed in all mCRC populations treated with surgery independently of the location of the primary tumour (Figure 1d). However, the population with mCRC originating in the right colon showed the most significant differences in the fractions of circulating *RAS* mutations between patients treated and patients not treated with tumour resection (Figure 1d). We observed that those patients diagnosed with synchronous mCRC originating in the right colon who had not yet received any treatment showed the highest ctDNA *RAS* MAF medians (Figure 1c). Conversely, the population with mCRC originating in the right colon treated with resection of the tumour contained the highest number of patients carrying plasmatic *RAS* MAFs closer to the 0.01% limit of detection of the BEAMing technology (Appendix A). Among them, patients with metachronous mCRC originating in the right side of the colon showed the lowest *RAS* MAF medians (Figure 1d), coinciding with the lowest sensitivity ratios and NPVs (Table 4).

## 4. Discussion

The mutational status of *RAS* genes is a well-known prognostic and predictive mCRC biomarker conferring poor prognosis and a lack of response to anti-*EGFR* antibodies. Therefore, guidelines strongly recommend the evaluation of the *RAS* mutational status before the choice of the first line of therapy [44,45,46,47]. Alternatively to the invasive SoC analyses carried out in locally restricted tumour tissue specimens obtained by surgery or throughout a biopsy at the time of the diagnosis of the disease, the evaluation of the *RAS* mutational status using plasmatic ctDNA isolated from peripheral blood allows a spatial-temporal view of the tumour [32]. Therefore, ctDNA assays have emerged as a promising minimally invasive tool with the potential to improve the management and therapy outcomes of the CRC patient by guiding clinical decisions [40,48].

Actually, ctDNA genotyping technologies and, more in particular, the OncoBEAM *RAS* CRC test, have demonstrated to be highly accurate tools, detecting down to 0.01% *RAS* allelic fractions in plasma samples from mCRC patients [32,40]. Hence, the detection of somatic *RAS* mutations through the analysis of plasmatic ctDNA in mCRC has been shown to be reliable and highly concordant with that observed in tumour tissue samples analysed with the gold standard tests [32,33]. In our study, using the mentioned technology to genotype plasmatic ctDNA from a real-world and heterogenous cohort of 366 mCRC patients, representative of the day-to-day care at four different hospitals in Galicia, we observed a concordance of 84.4% with the gold standard approach (SoC analysis in tumour tissue specimens). Similar to previous studies [32,49,50], the concordance ratios in our study were significantly higher in patients with liver metastases (89.7%) than in patients with metastases at other sites (76.1%), and in patients who had not been treated by resection of the primary tumour, as those with newly diagnosed synchronous tumours (90.2%), compared to those patients with synchronous (86.2%) and metachronous metastases (75.9%) who had already been treated by resection of the primary tumour.

In addition, binary logistic regression models, used to explore the odds of presenting discordant versus concordant assays depending on the mCRC clinicopathological characteristics, showed that the origin of the primary tumour was also an important predictor of discordant outcomes between the SoC and ctDNA tests. In fact, we observed that more than 91% of ctDNA genotypes from patients with mCRC originated in the left colon concorded with SoC tests. However, ctDNA tests carried out in samples from patients with mCRC originated in the rectum or right colon assigned the same genotype as SoC analysis in 84% and 77% of the cases, respectively. Up to 80% of the genotyping discrepancies observed in mCRC originated in the right colon and 63% of the discrepancies observed in mCRC originated in the rectum resulted from the failure of the ctDNA assay to detect any mutated *RAS* allele in the plasma of mCRC patients. 

The accuracy of the liquid biopsy tests to detect ctDNA isolated from peripheral blood depends on the fraction of mutated DNA within the whole cell-free DNA released to the bloodstream, mainly through cell death processes [51,52]. Although the characteristics of tumours known to be highly cfDNA shedders are still poorly understood, the anatomical location of the tumour lesions has been demonstrated to be an important factor determining the presence and, therefore, the detection of ctDNA in the plasma of patients with mCRC and other tumours [50,53,54]. In line with these observations, it has been proposed that the good vascularisation [40] and usually high volume of liver metastatic lesions [41,49] would contribute to increased cfDNA shedding into the bloodstream and, therefore, explain why mCRC patients diagnosed with liver metastases are frequently associated with higher plasmatic *RAS* MAFs, as we have also reported here. 

In this study, the mCRC subpopulation with at least one metastatic lesion in the liver showed significantly higher levels of cfDNA in plasma than patients without liver metastases but with metastases located at other anatomical sites. Consistent with this observation, the OncoBEAM assay detected higher fractions of mutated *RAS* alleles in plasma samples of patients with at least one liver metastasis than in patients without liver metastases. In addition, our results showed that the accuracy of the ctDNA assays was dramatically reduced to almost 50% when plasma samples belonged to mCRC cases with only metastases at the peritoneum, and to approximately 60% if samples had been obtained from cases with only lung metastases.

These new data obtained in a real-world cohort are in line with previous works addressed by ddPCR assays and also NGS and represent new evidence about the already known importance of the presence of liver metastases [32,37,41] to assess the plasmatic fractions of *RAS* mutations with enough sensitivity in mCRC patients.

Importantly, the plasmatic fraction of *RAS* mutations has also been shown to decrease after surgical resection of the primary tumour [34,36,40]. In agreement with these previous findings, patients from our cohort treated with resection of the primary tumour tissue showed significantly lower *RAS* MAFs in plasma than non-operated patients, with the lowest plasmatic ctDNA content observed in metachronous mCRC originating in the right colon and in the rectum. Consistent with the results, these populations showed negative predicted values below 50% and, thus, the highest probabilities of showing false-negative *RAS* genotypes in our study. 

Hence, our study also highlighted the relevance of the primary tumour location to understanding our capacity to detect plasmatic ctDNA in mCRC. 

It is well-accepted among the clinical and scientific community that the anatomical location of the primary tumour precedes the anatomical spreading of the metastatic lesions, with cancers originating in different bowel sites predisposing to different metastatic patterns [8]. Thus, left-sided mCRC, known to spread through the hepatic portal system to distal sites, is more prone to metastasis first to the liver and then spreading to the lungs. However, metastases in thoracic organs such as the lungs have more commonly been observed in rectal cancers with only a single metastatic site than in left colon cancers with single metastases [55]. On the other hand, patients with right-sided mCRC show peritoneal metastases with a higher frequency than patients with left-sided or rectal mCRC [8,56].

Therefore, in relation to the association of concordance ratios with the origin of the primary tumour in mCRC, we summarized that the higher ratios of concordance observed among assays from patients with primary tumours located at the left colon could be linked to the fact that these patients present clinicopathological features that have been significantly associated with higher fractions of mutated *RAS* in plasmatic ctDNA. Thus, more than 40% of these patients had more than one metastatic site when they were diagnosed with mCRC (Figure 1f), and as we have mentioned above, patients with mCRC that originated in the left colon have a higher preference to develop liver metastases [8]. Moreover, while the lung and the peritoneum were also organs that commonly showed metastases in patients with rectal and right-sided mCRC with only one metastatic site, the liver was the most common site for metastases in patients with mCRC originating in the left colon with only one metastatic site (Figure 1e). 

Based on these observations, left-sided mCRC has to be expected as a disease with increased ctDNA shedding and in general, presenting highly concordant *RAS* genotypes between tissue and plasma assays.

The association of the mCRC origin with the sensitivity to detect plasmatic fractions of mutated *RAS* alleles is of enormous relevance in the context of the selection of therapy for mCRC [16]. In a recent clinical trial, the evaluation of *RAS* mutations in plasma has already been shown to be relevant to monitor minimal residual disease after surgery in patients with non-metastatic CRC stages to decide the administration of adjuvant therapy [57]. Moreover, recent results obtained as part of the CIRCULATE study have shown that the detection of ctDNA after surgical treatment is more frequent in subjects with tumours located on the left side of the colon [58]. Therefore, based on our results, we advise that particular attention should be paid to mCRC originated in the right colon and rectum treated by resection, particularly in the metachronous setting, since the significantly reduced sensitivity observed in ctDNA assays from these patients could lead to inconclusive ctDNA results. In addition, the fact that 12% of the patients with WT *RAS* primary tumours originated in the rectum showed plasmatic *RAS* mutations when diagnosed with metachronous metastases later on is suggestive of the existence of molecular mechanisms of tumour evolution through the emergence of *RAS* mutations in these patients, which, if they are left inconclusively detected, could potentially reduce therapy responses.

Our study has some limitations, such as the different timing of the liquid biopsy collection, hindering the homogeneity of the population and making decisions regarding the data obtained. However, the observed post hoc statistical power based on the large sample size was high, and the BEAMing ddPCR genotyping was carried out in the same laboratory and by the same personnel, which makes the results reliable and reproducible.

## 5. Conclusions

In conclusion, our study confirmed that fractions of circulating *RAS* mutations and concordance between SoC and ctDNA assays were higher in mCRC with liver metastases and when the primary tumour was not resected. 

On the contrary, patients with metachronous mCRC that originated in the right colon or rectum, usually treated with resection of the primary tumour and presenting higher frequencies of single metastases at the peritoneum or the lung, showed the lowest fractions of circulating *RAS* mutations. Despite the high sensitivity of the BEAMing ddPCR technology, the evaluation of plasma circulating *RAS* mutations in patients with mCRC originating in the right colon or rectum, when operated on, showed a high probability of reporting false-negative outcomes and discordances with the SoC procedures. Approximately only 50% of the negative ctDNA tests in this mCRC population were predicted to be true negative and therefore showed a conclusive non-mutated *RAS* result.

## Figures and Tables

**Figure 1 cancers-15-03578-f001:**
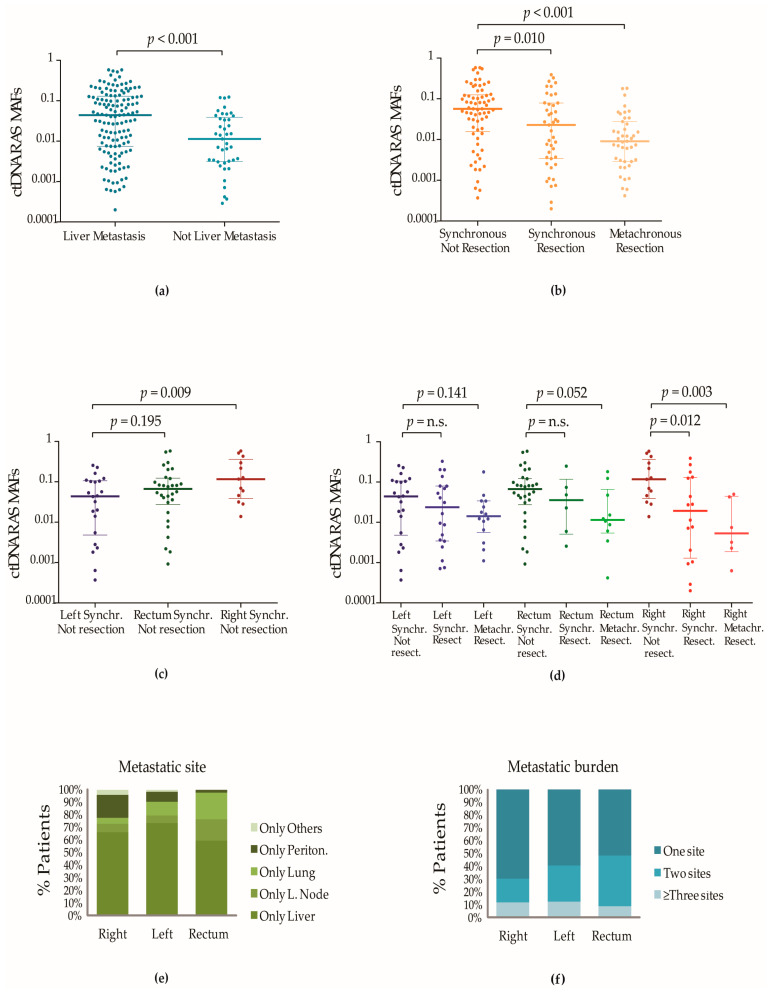
Impact of the mCRC clinicopathological characteristics on the detection of plasmatic *RAS* mutations. (**a**) Impact of the metastatic site on the plasmatic *RAS* MAFs. Dark blue dots (column 1) represent ctDNA *RAS* MAFs of patients with at least one liver metastasis. Light blue dots (column 2) represent ctDNA *RAS* MAFs of patients with at least one metastatic site at other anatomical locations than the liver. Lines represent the median and the 25–75% IQR. (**b**) Impact of surgical treatment and palliative/life-extending systemic chemotherapy on the ctDNA *RAS* MAFs. Plasmatic *RAS* mutations were detected in 113 patients with synchronous metastases and chemotherapy-naive and in 43 patients with metachronous metastases who had been at least previously treated by tumour surgery. Dark orange dots (column 1) represent ctDNA *RAS* MAFs of 69 patients with newly diagnosed synchronous mCRC who had not received surgical treatment. Light orange dots (column 2) represent ctDNA *RAS* MAFs of patients with synchronous metastases treated with resection of the primary tumour (4 had also been subjected to the removal of metastatic tissue). Yellow dots represent (column 3) ctDNA *RAS* MAFs of 43 patients with metachronous metastases; 34 had only been treated with resection of the primary tumour and 9 had been treated with resection of primary and metastatic tumour tissue. Adjuvant systemic chemotherapy had been administered to 24 of the cases before plasma collection. Lines represent the median and the 25–75% IQR. (**c**) Plasmatic *RAS* MAFs in patients with synchronous mCRC and different origins of the primary tumour who were not subjected to previous tumour surgery or systemic chemotherapy. Blue, green and red dots represent ctDNA *RAS* MAFs of patients with mCRC that originated in the left colon, rectum and right colon, respectively. Lines represent the median and the 25–75% IQR. (**d**) Plasmatic *RAS* MAFs from patients with primary tumours in different locations and treated with surgical resection. Blue, green and red dots represent ctDNA *RAS* MAFs of patients with mCRC that originated in the left colon, rectum and right colon, respectively. Colour gradients were used to show the decrease in the *RAS* MAFs in the mCRC subpopulations treated with resection of the primary tumour. Lines represent the median and the 25–75% IQR. (**e**) Correlation of primary tumour location with the metastatic pattern of mCRC. Plots showed the number of metastatic sites (**e**) and the anatomical location of the metastases (**f**) in 242 patients with synchronous tumours. All patients were naive to surgery or chemotherapy. The frequency of only one single site of metastasis was higher in patients with primary tumours located in the right colon. The liver was the predominant site of metastasis in mCRC with left colon origin, whereas lung metastases were more frequent in mCRC originating at the rectum. Peritoneal metastases and metastases to other sites were frequent in right-sided mCRC. Non-parametric Mann–Whitney U and Kruskal–Wallis tests were used to calculate *p*-values when two or more groups were compared.

**Table 1 cancers-15-03578-t001:** Clinicopathological characteristics of the mCRC patients included in the study.

Characteristics	Number	Percentage
Gender		
Female	104	28.4%
Male	262	71.6%
Age (years)		
Median (range)	67 (22–88)	
<50	29	7.9%
50–80	305	83.4%
>80	32	8.7%
Histology		
Adenocarcinoma	338	92.3%
Mucinous adenocarcinoma	14	3.8%
Others	4	1.1%
Unknown	10	2.7%
Primary tumour site		
Right (cecum, ascending and transverse colon)	79	21.6%
Left (descending and sigmoid colon)	154	42.1%
Rectum	112	30.6%
Mix	19	5.2%
Unknown	2	0.6%
Stage at diagnosis ^a^		
I	5	1.4%
II	32	8.7%
III	70	19.1%
IV	256	70.0%
Unknown	3	0.8%
Metastases presentation		
Metachronous	107	29.2%
Synchronous	256	70.0%
Unknown	3	0.8%
Number Metastatic sites		
1	204	55.7%
2	103	28.1%
≥3	50	13.7%
Unknown	9	2.5%
Metastases site		
Liver (only liver)	250 (125)	68.3% (34.2%)
Lung (only lung)	129 (29)	35.3% (7.9%)
Peritoneal (only peritoneal)	75 (27)	20.5% (7.4%)
Lymph node (only lymph nodes)	78 (15)	21.3% (4.1%)
Others (only others)	21 (7)	5.7% (1.9%)
Unknown	9	2.5%
Resection of primary tumour ^b^		
Yes	220	60.1%
No	144	39.3%
Unknown	2	0.6%
Resection metastases ^c^		
Yes	44	12.0%
No	320	87.4%
Unknown	2	0.6%
Systemic chemotherapy ^d^		
Yes	106	29.0%
No	254	69.4%
Unknown	6	1.6%
Anti-EGFR therapy ^e^		
Yes	122	33.3%
No	231	63.1%
Unknown	13	3.6%
Tissue biopsy location		
Primary tumour	256	69.9%
Metastasis	28	7.7%
Both	12	3.3%
Unknown	70	19.1%
Plasma ctDNA/BEAMing ^f^		
Mutated	180	49.2%
Non mutated	184	50.3%
Unknown	2	0.5%
Tumour Tissue/SoC ^g^		
Mutated	173	47.3%
Wild type	130	35.5%
Unknown	63	17.2%

Table 1 contains the clinical and pathological information of the 366 patients and samples recruited for the study. ^a^ Stage at the first diagnosis of CRC disease. ^b,c^ Surgery performed previous to plasma collection. ^d^ Systemic treatment received previous to plasma collection. ^e^ Anti-EGFR therapy received after plasma collection at any line of treatment. ^f^ A total of 55% (166/303) of the *RAS* mutations of the 303 tissue samples successfully genotyped with the SoC techniques were found in *KRAS* gene, and 3% (7/303) were found in *NRAS* gene. ^g^ BEAMing analysis showed *KRAS* mutations in 45% (165/366) of plasma samples and *NRAS* mutations in 5% (18/366).

**Table 2 cancers-15-03578-t002:** Concordance of *RAS* mutational status between SoC analysis in tumour tissue samples and ctDNA analysis using BEAMing technology.

	Nº	Concordance ^1^	Sensitivity ^2^	Specificity ^3^	K (95% CI) ^4^	OR (95% CI) ^5^	*p*-Value ^6^
All samples ^a^	301	84.4	81.3	88.5	0.69 (0.61–0.77)		
Primary tumour site ^b^							
Left Colon	126	91.3	87.1	95.3	0.83 (0.73–0.92)		
Rectum	97	83.5	82.5	85.0	0.66 (0.51–0.82)	2.1 (0.9–4.8)	0.083
Right Colon	65	76.9	73.3	85.0	0.52 (0.31–0.73)	3.1 (1.4–7.5)	0.008
Primary tumour resection ^c^							
Not	122	90.2	90.9	89.3	0.80 (0.70–0.91)		
Yes	177	81.4	75.2	90.3	0.63 (0.52–0.74)	1.9 (1.0–3.9)	0.073
Synchronous	94	86.2	81.0	94.4	0.72 (0.58–0.86)	1.5 (0.6–3.4)	n.s.
Metachronous	83	75.9	68.1	86.1	0.53 (0.34–0.71)	2.9 (0.4–6.5)	0.007
Metastatic site ^d^							
Liver							
Yes	204	89.7	90.2	89.0	0.79 (0.71–0.88)		
Not	88	76.1	65.5	93.9	0.53 (0.34–0.71)	2.7 (1.4–5.3)	0.003
L. nodes	65	86.2	86.8	85.2	0.72 (0.54–0.89)		
Yes							
Not	227	85.5	80.8	91.8	0.71 (0.62–0.80)	1.1 (0.5–2.5)	n.s.
Lung							
Yes	110	85.5	82.9	90.0	0.70 (0.56–0.84)		
Not	182	85.7	81.6	90.5	0.72 (0.61–0.82)	1.0 (0.5–2.0)	n.s.
Peritoneum							
Yes	61	82.0	71.9	93.1	0.64 (0.45–0.84)		
Not	231	86.6	84.6	89.5	0.72 (0.63–0.81)	0.7 (0.3–1.6)	n.s.
Nº metastatic sites ^e^							
Multiple	128	89.1	89.0	89.1	0.78 (0.67–0.89)		
Single	164	82.9	76.8	91.3	0.66 (0.55–0.78)	1.7 (0.9–3.4)	0.141
Liver	98	88.8	88.5	89.1	0.78 (0.65–0.90)	1.0 (0.4–2.4)	n.s.
L. nodes	12	83.3	85.7	80.0	0.67 (0.22–1.00)	1.6 (0.2–7.0)	n.s.
Lung	25	72.0	63.2	100.0	0.45 (0.11–0.80	3.2 (1.1–8.8)	0.029
Peritoneum	21	71.4	53.8	100.0	0.47 (0.11–0.83)	3.3 (1.0–9.5)	0.035
Others	7	71.4	50.0	100.0	0.46 (0.00–1.00)	3.3 (0.4–16.8)	0.181

Table 2 includes information on 301 patients with available *RAS* status in solid tumour tissue and plasmatic ctDNA samples. ^1^ Overall percentage agreement; ^2^ positive percentage agreement; ^3^ negative percentage agreement; ^4^ Cohen’s kappa coefficient (95% confidence interval); ^5^ odds ratio (95% confidence interval); ^6^ n.s. = *p*-value > 0.05. ^a^ Discordant outcomes between SoC and BEAMing analyses were observed in 47 (15.6%) of 301 patients with *RAS* status available in tumour tissue and in plasma ctDNA samples: 10.6% of cases showed exclusive *RAS* mutations in the tumour tissue sample, and in 5% of cases, *RAS* was found to be mutated in the ctDNA sample of a patient with a wild-type *RAS* genotype in the tumour tissue sample. ^b^ Information about the location of the primary tumour was available in 288 patients. Patients with mixed locations were excluded from this analysis. Left Colon refers to primary tumours located at the descending and sigmoid colon. Right Colon refers to primary tumours located at the cecum or the ascending and transverse colon. All ORs were calculated in reference to the left colon location. ^c^ Information about surgical resection of tumour tissue was available in 299 cases. The time of presentation of the metastatic disease was known in 299 patients. Not, patients diagnosed with synchronous mCRC who had not undergone surgery or any treatment before the collection of the liquid biopsy sample. Yes, patients who had been treated with resection of the primary tumour before the collection of the liquid biopsy sample (33 patients had also been treated with resection of metastatic tissue). Synchronous refers to cases diagnosed with synchronous mCRC who had undergone resection of the primary tumour before the collection of the liquid biopsy sample (17 patients had also been treated with resection of metastatic tissue). All patients with metachronous metastases had undergone resection of the primary tumour (16 patients had also been treated with resection of metastatic tissue). ORs were calculated in reference to cases with no resection. ^d^ Site of metastases. Numbers included mCRC cases with (Yes) or without (Not) the metastases at the indicated site. ORs were referred to cases with the specified metastatic site. ^e^ Number of metastatic sites was obtained in 292 cases. Multiple sites include all cases showing metastases in more than one anatomical location. All ORs were calculated in reference to cases with multiple metastatic sites. n.s. indicates not significant *p*-values.

**Table 3 cancers-15-03578-t003:** Discordance of *RAS* mutational status between SoC and ctDNA analyses associated with the mCRC origin site.

	Left Colon			Rectum			Right Colon		
	Nº	OR (95% CI)	*p*-Value	Nº	OR (95% CI)	*p*-Value	Nº	OR (95% CI)	*p*-Value
Primary tumour resection ^a^									
Not	40			57			19		
Yes	86	1.3 (0.3–6.0)	n.s.	40	5.7 (1.8–21.8)	0.005	46	1.9 (0.5–9.1)	n.s.
Synchronous	47	1.5 (0.3–7.5)	n.s.	15	2.0 (0.3–11.7)	n.s.	29	1.4 (0.3–7.4)	n.s.
Metachronous	39	1.0 (0.2–5.9)	n.s.	25	8.8 (2.6–36.1)	0.001	17	2.9 (0.6–16.2)	0.187
Metastatic site ^b^									
Liver									
Yes	90			63			42		
Not	36	0.9 (0.2–3.5)	n.s.	30	5.0 (1.5–17.8)	0.009	20	6.1 (1.7–26.5)	0.006
L. Nodes									
Yes	21			23			18		
Not	105	0.9 (0.2–6.2)	n.s.	70	2.2 (0.5–14.7)	n.s.	44	1.0 (0.3–4.3)	n.s.
Lung									
Yes	41			53			14		
Not	85	1.3 (0.4–6.3)	n.s.	40	0.7 (0.2–2.2)	n.s.	48	1.1 (0.3–5.4)	0.046
Peritoneum									
Yes	32			6			20		
Not	94	1.6 (0.4–10.8)	n.s.	87	0.2 (0.0–0.9)	0.027	42	0.8 (0.2–3.0)	n.s.
Nº metastatic sites ^c^									
Multiple	47			51			26		
Single	79	1.1 (0.3–4.2)	n.s.	42	2.5 (0.8–8.8)	0.127	36	3.4 (0.9–16.3)	0.088
Liver	49	1.0 (0.2–4.3)	n.s.	22	0.9 (0.1–4.6)	n.s.	23	1.6 (0.3–9.1)	n.s.
L. nodes	6	2.2 (0.1–18.7)	n.s.	3	n.a.	n.a.	2	7.7 (0.3–234)	0.186
Lung	10	n.a.	n.a.	13	5.8 (1.3–2.6)	0.018	2	n.a.	n.a.
Peritoneum	11	1.1 (0.1–8.3)	n.s.	1	n.a.	n.a.	8	7.7 (1.3–54.5)	0.030
Others	3	5.4 (0.2–70.4)	n.s.	3	4.6 (0.2–57.5)	n.s.	1	n.a.	n.s

Table 3 includes information from 288 patients with known clinicopathological data. The location of the primary tumour was unknown in 13 of the 301 patients with available *RAS* status in solid tumour tissue and plasmatic ctDNA samples. ^a^ Information on the location of the primary tumour was available in 288 patients. Patients with mixed locations were excluded from this analysis. ORs were calculated in reference to cases with no resection. ^b^ The site of metastases was available in 281 patients. The numbers included mCRC cases with (Yes) or without (Not) metastases at the indicated anatomical site. ORs were referred to cases with the specified metastatic site. ^c^ The number of metastatic sites was obtained in 281 cases. Multiple sites included all cases with metastases in more than one anatomical location. All ORs were calculated in reference to cases with multiple metastatic sites. n.a., statistical power was not enough for calculation. n.s, *not significant p*-values.

**Table 4 cancers-15-03578-t004:** Concordance of *RAS* mutational status between SoC and ctDNA analyses related to tumour surgery.

	Nº	Concordance ^1^	Sensitivity ^2^	Specificity ^3^	K ^4^ (95% CI)	PPV ^5^ (95% CI)	NPV ^6^ (95% CI)
Synchronous not resection							
Left	40	92.5	90.0	95.0	0.85 (0.69–1.00)	97.7 (71.9–99.7)	90.5 (68.2–98.3)
Rectum	57	93.0	96.7	88.9	0.86 (0.73–0.99)	90.6 (73.8–97.5)	96.0 (77.7–99.8)
Right	19	84.2	90.9	75.0	0.67 (0.33–1.00)	83.3 (50.8–97.1)	75.0 (35.6–95.5)
Synchronous resection ^a^							
Left	47	89.4	86.4	92.0	0.79 (0.61–0.96)	90.5 (68.2–98.3)	88.5 (68.7–97.0)
Rectum	15	86.7	81.8	100.0	0.71 (0.33–1.00)	100.0 (62.9–100.0)	66.7 (24.1–94.0)
Right	29	79.3	73.9	100.0	0.54 (0.21–0.87)	100.0 (77.1–100.0)	50.0 (22.3–77.7)
Metachronous ^b^							
Left	39	92.3	85.0	100.0	0.85 (0.68–1.0)	100.0 (77.1–100.0)	86.4 (64.0–96.4)
Rectum	25	60.0	56.3	66.7	0.21 (0.0–0.59)	75.0 (42.8–93.3)	46.2 (20.4–73.9)
Right	17	64.7	54.6	83.3	0.33 (0.0–0.76)	85.7 (42.0–99.2)	50.0 (20.1–79.9)

Table 4 includes information from 288 patients with known clinicopathological data. The location of the primary tumour was unknown in 13 of the total 301 patients with available *RAS* status in solid tumour tissue and plasmatic ctDNA samples. Discordant outcomes between SoC and BEAMing analyses were observed in 23 cases (12.1%) with synchronous metastases and 19 cases (24.7%) with metachronous metastases. ^1^ Overall percentage agreement; ^2^ positive percentage agreement; ^3^ negative percentage agreement; ^4^ Cohen’s kappa coefficient (95% confidence interval); ^5^ positive predictive value (95% confidence interval); ^6^ negative predictive value (95% confidence interval). ^a^ Primary tumour resection was performed in 43.5% of patients with synchronous metastases. ^b^ Primary tumour resection was performed in 100% of patients with metachronous metastases.

## Data Availability

The data presented in this study are available on request from the corresponding authors.

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
