# Peer review of "Clinicopathological Profiles Associated with Discordant RAS Mutational Status between Liquid and Tissue Biopsies in a Real-World Cohort of Metastatic Colorectal Cancer"

_cancers, 2023, doi:10.3390/cancers15143578_

Round 1
Reviewer 1 Report
Very good article
Author Response
I would like to thank the reviewer for reading the manuscript and send the report and comments.
Reviewer 2 Report
From a biostats/clinical epidemiology point of view, I have some suggestions for the Authors:
- line 39 correlated: better to say associated, here and everywhere (don't forget you are dealing with Cohen's kappa for categorical covariates! you're not comparing continuous covariates!)
- table 1, pTNM, cTNM and Dukes staging infos could be of help for the clinical oncologists
- table 1, 30 pts with unknown age!? these missing data have to be solved!
- line 161, could you better define main outcome, i.e. the odds of discordance versus the odds of non discordance (if I have well understood!)?
- descriptive stats for continuous variables are to be reported as median/IQR, thus avoiding mean/sd
- p-values always reported with 3 sign-digits
- at the light of the above comment, a non-parametric inferential approach is strongly advised
- tables 2-3-4, too messy, better to shorten them
- the role of binary logistic regression model has to be more properly described: remember that this model does not estimate discordance!
- conclusion, a bit tricky, why not to add a more formal take-home message!?
minor
Author Response
- line 39 correlated: better to say associated, here and everywhere (don't forget you are dealing with Cohen's kappa for categorical covariates! you're not comparing continuous covariates!)
Line 33: “Correlated” has been changed by “associated”.
- table 1, pTNM, cTNM and Dukes staging infos could be of help for the clinical oncologists
Staging at the diagnosis has now been included in the Table 1
|
Stage at diagnosis a |
|
|
|
I |
5 |
1.4% |
|
II |
32 |
8.7% |
|
III |
70 |
19.1% |
|
IV |
256 |
70.0% |
|
Unknown |
3 |
0.8% |
a Stage at the first diagnosis of CRC disease.
In addition, an explanatory text for the classification has been written at the Methods and Materials section:
Line 127: “All patients had been previously diagnosed of CRC Stage I, Stage II (IIA, IIB, IIC), Stage III (IIIA, IIIB, IIIC) or Stage IV (IVA, IVB ad IVC) following the recommendations of the American Joint Committee on Cancer (AJCC) TNM Staging System for Colon Cancer [42] published at the NCCN Guidelines for Colon Cancer [43] . All patients were selected to have been histological and radiological diagnosed of Stage IV metastatic colorectal cancer at the time of the enrolment in this study.”
- table 1, 30 pts with unknown age!? these missing data have to be solved!
The age of all subjects has now been reported in the table 1
|
Median (range) |
67 (22-88) |
|
|
<50 |
29 |
7.9% |
|
50-80 |
305 |
83.4% |
|
>80 |
32 |
8.7% |
- line 161, could you better define main outcome, i.e. the odds of discordance versus the odds of non discordance (if I have well understood!)?
This point has now been defined in the methods:
Line 167-168: “The main outcome of the analysis was the odds of discordance versus concordance between SoC and ctDNA tests. “
- descriptive stats for continuous variables are to be reported as median/IQR, thus avoiding mean/sd
Thanks for your considerations regarding this point. We would like to clarify that the previous text was already showing the median for each statistical description mentioned. However, a range from the min and the max levels of the values was highlighted instead of the IQR. We appreciated your counselling and we have now modified the text to include the IQR at all places of the statistics.
Line 291-292: “Median of RAS MAFs in plasmatic ctDNA from 165 patients with RAS mutated genotype was 0.029, IQR [0.005-0.104], ranging from a minimun level of 0.0002 to a maximum of 0.58.”
Line 296: “…(median=0.044, IQR [0.007-0.129] vs. median=0.011, IQR [0.003-0.039]; p-value<0.001) (Figure 1A).”
Line 300: "…(median=36.7 ng/ml, IQR [17.7-87.3] vs. median=12.9 ng/ml, IQR [9.8-18.9]; p-value < 0.001).”
Line 302-304: “….with lung metastases alone (median=37.1 ng/ml, IQR [20.4-80.4] vs. 13.2 ng/ml, IQR [12.5-17.2], p-value = 0.004) or exclusive peritoneal metastatic disease (median=37.1 ng/ml, IQR [20.4-80.4] vs. 12.4 ng/ml, IQR [9.8-20.5], p-value = 0.031).
- p-values always reported with 3 sign-digits.
Regarding to this point, I have now modified the text and figure 1 to include 3 sign-digits at all places:
Line 245: “the blood liquid biopsy (OR=1.9, 95% CI [1.0 - 3.9]; p-value= 0.073)….”
Line 258-: “…..the left colon (OR=2.1, 95% CI [0.9 - 4.8]; P-value= 0.083) (Table 2).”
Line 300: ” (median=36.7 ng/ml, IQR [17.7-87.3] vs. median=12.9 ng/ml, IQR [9.8-18.9]; p-value < 0.001).”
Line306-307: “(median=37.1 ng/ml, IQR [20.4-80.4] vs. 13.2 ng/ml, IQR [12.5-17.2], p-value = 0.004)”
Line307-308: “(median=37.1 ng/ml, IQR [20.4-80.4] vs. 12.4 ng/ml, IQR [9.8-20.5], p-value = 0.031)”
Line 313-314: “(median= 0.057, IQR [0.016-0.125] vs. median= 0.025, IQR [0.003-0.079]; p-value = 0.010)”
Line 311-312:” ….metachronous metastases (median= 0.009, IQR [0.003-0.028] ; p-value <0.001 ).”
Figures 1a, 1b, 1c and 1d now containing p-values with 3 sign-digits
- at the light of the above comment, a non-parametric inferential approach is strongly advised
Non-parametric Mann-Whitney and Kruskal Wallis tests had already been carried out for the analyses however to avoid any doubts, the next paragraph was included in the methods:
Line 183-189: “The median and interquartile range (25%-75%; IQR) of the cfDNA and MAFs levels distributions were calculated for the different mCRC subpopulations. Non-parametric Mann-Whitney and Kruskal Wallis approaches were applied for statistical comparisons when two or more populations were compared”.
In addition, the next paragraph has been included in the Figure1 notes:
Line 329, line 338, line 341, line 343: “Lines represent the median and the 25%-75% IQR.”
Line 350: “Non-parametric Mann-Whitney U and Kruskal-Wallis tests were used to calculate p-values when two or more than two groups were compared.”
- tables 2-3-4, too messy, better to shorten them
Tables 2-3-4 have been modified to simplified the content:
- The 95% CI of the Concordance, Specificity and Sensitivity values have been eliminated.
- Numbers reporting the counts of TN, TP, FN and FP have been eliminated and only the total number of patients per group has been left in the tables.
- p-values > 0.05 have been written as n.s. (not significant)
- Some of the Titles have also been shortened
- the role of binary logistic regression model has to be more properly described: remember that this model does not estimate discordance!
The role of the binary logistic regression models has now been addressed in the Material and Methods within the Statistical analysis Section, together with the explanation of the main outcome, by including the following paragraph:
Line 160: “Statistical analysis were performed with clinically predefined mCRC subpopulations.
Line 165: “Binary logistic regression was used to model the relationship between mCRC clinicopathological characteristics and the RAS mutational status (discordant vs. concordant) taken as the dichotomous dependent variable. The main outcome of the analysis was the odds of discordance versus concordance between SoC and ctDNA tests. We used a generalized linear model and a logit function to estimate the Odd Ratios (ORs) of discordance and their 95% confidence intervals (CI); and to calculate the probability of association with the primary tumour site, the primary tumour resection, the metastatic site and the number of metastatic sites, taken as the independent variables. Patients with mCRC of left colon origin, not primary tumour resection, with liver, lymph nodes, lung or peritoneal metastases, and multiple metastatic sites were used as the reference for the independent variables.
Line 175: We used univariable logistic regression to estimate the effect of each of the clinicopathological characteristics on the odds of discordance occurring, individually. Multivariable logistic regression models were also used to estimate the jointly effect or interaction of a set of clinicopathological variables on the probability of the outcome happening.
Line 179: Finally, univariable logistic regression analysis in mCRC subpopulations stratified by the primary tumour site (Left colon, Rectum and Right colon) were undertook to model the effect of the primary tumour resection, the metastatic site and the number of metastatic sites on the odds of discordance occurring as described in table 4.”
In addition, I have introduced some text on the discussion to clarify the role of the binary logistic regression model which I hope could clarify this question:
Line 377: “In addition, binary logistic regression models, used to explore the odds of presenting discordant vs. concordant assays depending on the mCRC clinicopathological characteristics, showed that origin of the primary tumour was also an important predictor of discordant outcomes between SoC and ctDNA tests.”
- conclusion, a bit tricky, why not to add a more formal take-home message!?
We appreciate the reviewer comment. Our previous conclusions aimed to give the readers a clear message about in which cases a negative test assessing RAS status on cfDNA should be interpreted with caution and consider a tissue biopsy or ctDNA close monitoring. However, we have now modified the conclusion describing the main results and formal conclusions obtained in the study:
Line 461-471: “In conclusion, our study confirmed that fractions of circulating RAS mutations and concordance between SoC and ctDNA assays were higher in mCRC with liver metastases and when the primary tumour was not resected.
On the contrary, patients with metachronous mCRC originated at the right colon or rectum, usually treated by tumour resection and with a higher frequency of single metastases at the peritoneum or lungs, showed the lowest fractions of circulating RAS mutations. Despite the high sensitivity of the BEAMing ddPCR technology, the evaluation of plasma circulating RAS mutations in mCRC originated at the right colon or rectum, when operated, showed a high probability of reporting false negative outcomes and discordances with the SoC procedures. Approximately only 50% of the negative ctDNA tests in this mCRC population were predicted to be actually true negative and therefore showed a conclusive non-mutated RAS result.”

Reviewer 3 Report
First of all, I would like to congratulate the authors for this very interesting and original paper. ctDNA has become a valuable tool helping to tailor treatment in mCRC and RAS/BRAF/MMR status as well as sidedness are fundamental to select 1L in mCRC. Please find attached the draft with very few text edits recommendations.
Congrats!

Author Response
First of all, I would like to congratulate the authors for this very interesting and original paper. ctDNA has become a valuable tool helping to tailor treatment in mCRC and RAS/BRAF/MMR status as well as sidedness are fundamental to select 1L in mCRC. Please find attached the draft with very few text edits recommendations.
I appreciate the reviewer recommendations.
1. Regarding to the text edits, the proposed suggestions have now been introduced in the text at the following lines:
Line 78-80: “…have demonstrated to improve outcomes in mCRC patients with wild type (WT) status compared to anti-VEGF in the first line [9–11] …”
2. A citation for the prevalence of RAS mutations in CRC has also been included:
Line 81-82: “However, approximately 40% of the CRCs show activating missense mutations at the Kirsten RAS (KRAS) gene, and 3–5% at the neuroblastoma RAS (NRAS) [18],…..”
3. Regarding to the reviewer comment about “HRAS is not mandatory for CRC”, the text has been modified and left as following:
Line 84-86: “Hence, current guidelines for the clinical management of mCRC recommend the evaluation of RAS mutational status, especially exons 2, 3 and 4 of the KRAS and NRAS genes before the administration of EGFR inhibitors [20–22].”
Round 2
Reviewer 2 Report
I have no more questions for the Authors